# Low-Cost Three-Dimensional Modeling of Crop Plants

**DOI:** 10.3390/s19132883

**Published:** 2019-06-28

**Authors:** Jorge Martinez-Guanter, Ángela Ribeiro, Gerassimos G. Peteinatos, Manuel Pérez-Ruiz, Roland Gerhards, José María Bengochea-Guevara, Jannis Machleb, Dionisio Andújar

**Affiliations:** 1Department of Aerospace Engineering and Fluids Mechanics, Escuela Técnica Superior de Ingeniería Agronómica (ETSIA), Universidad de Sevilla, 41013 Sevilla, Spain; 2Centre for Automation and Robotics, CSIC-UPM, Arganda del Rey, 28500 Madrid, Spain; 3Department of Weed Science, Institute of Phytomedicine, University of Hohenheim, Otto-Sander-Straße 5, 70599 Stuttgart, Germany

**Keywords:** plant phenotyping, RGB-D, Structure from Motion, RGB-D

## Abstract

Plant modeling can provide a more detailed overview regarding the basis of plant development throughout the life cycle. Three-dimensional processing algorithms are rapidly expanding in plant phenotyping programmes and in decision-making for agronomic management. Several methods have already been tested, but for practical implementations the trade-off between equipment cost, computational resources needed and the fidelity and accuracy in the reconstruction of the end-details needs to be assessed and quantified. This study examined the suitability of two low-cost systems for plant reconstruction. A low-cost Structure from Motion (SfM) technique was used to create 3D models for plant crop reconstruction. In the second method, an acquisition and reconstruction algorithm using an RGB-Depth Kinect v2 sensor was tested following a similar image acquisition procedure. The information was processed to create a dense point cloud, which allowed the creation of a 3D-polygon mesh representing every scanned plant. The selected crop plants corresponded to three different crops (maize, sugar beet and sunflower) that have structural and biological differences. The parameters measured from the model were validated with ground truth data of plant height, leaf area index and plant dry biomass using regression methods. The results showed strong consistency with good correlations between the calculated values in the models and the ground truth information. Although, the values obtained were always accurately estimated, differences between the methods and among the crops were found. The SfM method showed a slightly better result with regard to the reconstruction the end-details and the accuracy of the height estimation. Although the use of the processing algorithm is relatively fast, the use of RGB-D information is faster during the creation of the 3D models. Thus, both methods demonstrated robust results and provided great potential for use in both for indoor and outdoor scenarios. Consequently, these low-cost systems for 3D modeling are suitable for several situations where there is a need for model generation and also provide a favourable time-cost relationship.

## 1. Introduction

Digital models allow information gathering about crop status for agricultural management or breeding programmess. The use of high-throughput plant phenotyping greatly impacts in plant characterization and plant selection processes [1]. Digital plant models can be objective tools for quantifying plant characteristics that avoid the appreciative differences incorporated in human judgement. Phenological traits can be accurately measured, expediting the decision-making processes [2]. Thus, plant models can measure and characterize complex plant shapes, providing essential information to plant breeding programmes that is necessary for modifying traits related to physiology, architecture, stress or agronomical management [3]. Plant modeling makes information, such as plant treatment or growth assessments, more accessible to agricultural managers, therefore providing managers with a detailed and comprehensive understanding of plant development throughout the life cycle. In addition to plant breeding, digital models can be used to help growers in disease detection, yield estimation or biomass production, to characterize fruit quality, discriminate between weeds and crop, to correlate with the photosynthetic activity, etc. Although the creation of digital models allows for a better understanding of the internal processes in plant growing, they require technological developments for sensing and capturing purposes. New sensors and procedures are required to achieve these objectives. Electronic devices can rapidly characterize crops in a non-destructive, accurate and repeatable manner. Most of the sensing technologies used are based on two-dimensional characterization, from visible imagery to thermal, multispectral imaging or fluorescence sensors [4]. Fluorescence sensors are typically used for the extraction of parameters related to leaf composition; they expose plants to a specific wavelength of visible or ultraviolet (UV) light, and the emitted fluorescent radiation is measured and the calculated values are related with nitrogen content, plant stress, etc., due to the metabolic changes occurring in the chloroplast. This method is typically used for measuring the response to induced stress [5]. In similar applications, spectral reflectance sensors are commonly used for nitrogen assessment or to separate plants from the ground by distinguishing the values of different spectral bands [6]. In this way, it is possible to combine these spectral values to quantify crop aspects such as leaf area index (LAI), biomass or expected yield. Thermal imaging is generally used for the monitoring of water stress by measuring the leaf or canopy temperature [7] whereby the measured values are related to the evapotranspiration capacity of the plants and the water availability, in order to manage precise irrigation systems. Visible imaging methods are the most extensively developed; they are widely used and have been improved in recent decades [8]. These methods are a fast, reliable and simple ways to describe plant dynamics using high resolution and low-cost cameras. However, some problems arise when these methods are used in complex environments, such as uncontrolled illumination, the presence of shadows or overlapping leaves leading to information loss, which may affect the accuracy of the measurements [9]. Although using red-green-blue (RGB) cameras is a common method of plant phenotyping, the details included in a planar image only provide a limited view of the scene. However, these cameras are widely used for phenology monitoring, nitrogen application, yield monitoring and weed discrimination [10]. 

The aforementioned sensors only provide information in a two-dimensional plane while other characteristics of the third dimension are hidden. The creation of 3D models by distance sensors is opening new opportunities in phenotyping processes by increasing efficiency and accuracy [11]. During the last two decades, the use of distance sensors has permitted distance measurement from the sensor to an object, thereby becoming widely used in agricultural research. Distance sensors can describe plant morphology with high accuracy and 3D models can be created by the addition of a third dimension due to the movement and spatial displacement of the sensor. The main distance sensors that are available for many operations and that can supply data suitable to create 3D models for plant reconstruction are ultrasonic and LiDAR (light detection and ranging) sensors. However, these two types of sensors are completely different with regards to their mechanism, cost, field of view and accuracy. An ultrasonic sensor measures distance an echo; sound waves are transmitted and return back to their source as an echo after striking an obstacle. Since the speed of the sound is known and the travelling time can be measured, the distance is then calculated. This methodology has been used for different agronomical tasks, such as crop characterization, fertilizer application or weed discrimination [12]. Although, the use of ultrasonic devices for canopy identification is commonly accepted as a good method for tree characterization and fertilization, their limited resolution in combination with their wide field of view makes them an invalid tool for precise plant description. On the other hand, LiDAR sensors are able to provide 2D- or 3D-plant models by displacing the sensor following a path and storing the sensor’s relative position. The LiDAR sensors fire rapid pulses of laser light at a surface and measure the amount of time it takes for each pulse to bounce back; this time is then converted to distance. The accuracy of LiDAR is higher than that of ultrasonic sensors, and they have been widely used for tree canopy description [13] and lastly in breeding programmes creating high-resolution 3D models. LiDAR sensors are robust and reliable and they allow scanning at high frequency and over large distances. Similar systems, such as radar systems [14], hemispherical photography [15], and magnetic resonance and X-ray visualization, have been studied for plant morphology characterization [16]. 

Although the aforementioned methods can accurately describe plant shape, they are time-consuming and lack colour information, which, in many cases, limits their application. While colour can be added to the scene by means of sensor fusion, this combination of the traditional sensing devices with RGB information from a camera is technically challenging and remains limited. The emergence of new sensors in the market coming from different sectors opened new modes of plant description. Initially distributed mainly by the gaming industry, today there are a large number of commercially available RGB-Depth cameras. The more advanced ones combine more than one sensor like the Multisense from Carnegie Robotics (Carnegie Robotics, LLC, Pittsburgh, Pennsylvania) which combines lasers, depth cameras and stereoscopic vision; the ZED Stereo Camera (Stereolabs Inc, San Francisco, CA, USA) which also provides depth information; or the DUO MLX sensor (Code Laboratories Inc, Henderson, NV, USA) which uses a stereoscopic camera to provide depth information. 

However, Microsoft Kinect v2® (Microsoft Corp., Redmond, WA, USA) is the primary and most widely known sensor in this market and has been widely used for plant characterization in agriculture. For instance, the maximum diameter, volume and density of sweet onions were estimated with an accuracy of 96% [17]. Paulus, et al. [18] evaluated the possibilities of RGB-D cameras in comparison with a high-precision laser scanner to measure the taproots of sugar beet, the leaves of sugar beets and the shape of wheat ears. These authors concluded that the low-cost sensors could replace an expensive laser scanner in many of the plant phenotyping scenarios. Andújar, et al. [19] reconstructed trees at different wind speeds to estimate the LAI and the tree volume; the resolution potentials of this sensor from null wind speed to high wind speeds were successfully demonstrated as the 3D models were properly reconstructed at high wind speeds. Far from being used only for plant characterization, a Kinect sensor could be used to estimate plant health. The NDVI (normalized difference vegetation index) can be calculated by using the values of near-infrared (NIR) pixels and red pixels for plant health monitoring [20]. Consequently, new applications for sensors, unlike those traditionally used in plant phenotyping, are emerging. However, there are still some unresolved issues, such as a proper reconstruction of the end-details required in some phenotyping programmess. In this regard, a new approach has emerged in recent years; the use of photogrammetry using low-cost RGB cameras has a high impact in the phenotyping processes. This method, also known as stereo vision, is relatively inexpensive but allows the acquisition of a high definition of details. By examining the relative positions of objects in the two planes similar to the human vision, 3D information can be extracted. The 3D scene can be reconstructed either by a single camera or multiple cameras shooting around the object of interest. For model reconstruction, some parameters should be fixed prior to image acquisition; for example, the distance from the camera to the target should be established based on the camera’s focal length in order to reach a proper overlap between images according to the reconstruction algorithm. This principle has been already used for plant reconstruction to extract information regarding plant height, LAI and leaf position [21]. In addition, a combination of cameras and images from Structure from Motion (SfM) creates high-fidelity models wherein a dense point cloud is created by means of a growing region [22]. Similarly, the multi-view stereo (MVS) technique creates dense 3D models by fusing images [23]. Nevertheless, there are still issues that should be improved and considered during the reconstruction processes. Boundaries and limitations of methods must be clearly stated for its usage. Some aspects that need to be improved, such as the accurate reconstruction of end-details, should be well know before of applying reconstruction techniques for more detailed applications. In addition, these low-cost methods for plant reconstruction are opening up possibilities for an easy use of a single camera or a budget device without expensive additional equipment for plant model reconstruction. 

Thus, research of new methods and modes that employ depth for 3D plant reconstruction for plant phenotyping has a strong impact and still are of great interest to the sector, in the search for inexpensive equipment that can perform field high-throughput phenotyping functions with adequate precision requirements. Furthermore, the use of 3D-structural models can improve the decision-making processes. The objective of this study was to assess the combination of some of the most novel methods and sensors for plant reconstruction and to compare them to the planar RGB method. The capacities of RGB-D cameras and photogrammetry for 3D model reconstruction of three crops at different stages of development were compared. The crops selected to be characterized had differential shapes; they were maize, sugar beet and sunflowers. We specifically explored the capabilities and limitations of the different principles and systems in terms of accuracy to extract various parameters related to the morphology of these crops. The methodology compared the accuracy and the capacities of self-develop algorithms to process the point clouds for solid model construction. 

## 2. Materials and Methods

### 2.1. Site Location and Modeling Systems

RGB-D cameras and photogrammetry methods were tested using three different crops with contrasting plant structures at a research station of the University of Hohenheim (Stuttgart, Germany). The crops maize (*Zea mays* L.) sugar beet (Beta vulgaris L.), and sunflower (*Helianthus annuus* L.) were selected for ascertaining the structural differences between them at three different stages of development. The stage of development for maize was ranged from 13 to 19 on the BBCH (Biologische Bundesanstalt, Bundessortenamt und Chemische Industrie) scale [24]. The stage of development for sunflower was ranged from 15 to 19 on the BBCH scale, and the stage of development for sugar beet ranged from 16 to 19 on the BBCH scale. Three samples per crop and stages of development were monitored. The plants were randomly selected from the field and ensured to be a representative sample of the population. Closer plants were removed to avoid interference with the target plant during monitoring. Although removing those plants does not affect the model reconstruction it facilitates to take the model measurements. To scale the model during post-processing, tree graphical scales were located at the ground level in a triangular shape around the plant stem, with the plant located in the centre of the triangle. 

### 2.2. Data Acquisition and Data Processing

#### 2.2.1. Low-Cost Photogrammetry

The SfM method was applied for model reconstruction. A set of images covering each crop plant was used and the data set was composed of approximately 30 to 40 images per sample to fully cover of the plant based on its size. The plant was manually photographed with a Sony Cyber-shot DSC-HX60 camera (Sony, Tokyo, Japan) following a concentric track at three different heights to produce baseline image pairs (Figure 1). The images were taken with a minimum overlapping with the previous image at 90%. The camera was manually held at distance approximately 50 cm from the plant stem in every shoot. The minimum number of images was set up by a previous study to determine the lowest number of images for a proper reconstruction. These images guarantee a proper reconstruction. An increase in the images did not provide any improvement in the accuracy and the extracted values did not show any statistical significance. The method and distance to the target objects was set by [23]. The positions of the camera were not previously established since the SfM algorithm can correct the small differences during the model creation. 

The model was processed in Agisoft PhotoScan Professional Edition software (Agisoft LLC, St. Petersburg, Russia) version 1.0.4. to build the digital surface model (DSM) of each plant. The camera position and the common points in the images were located and matched; this process refined the camera calibration parameters. Then, the DSM was constructed based on the relationship of the camera position to the images. The projection of the images and its combination allowed the creation of the DSM by searching for common points in the scene. The process created a dense 3D-point cloud, which was later used for the solid model creation. The processing was divided in three automated steps: (1) aligning images; (2) building field geometry; and (3) generating dense point clouds. Thereafter, the model was manually scaled according to the graphical scales located around the plant stem. 

#### 2.2.2. Red-Green-Blue (RGB) Image Processing 

The RGB images processed as planar images were acquired with the same camera (DSC-HX60V, Sony, Tokyo, Japan) used in the photogrammetric method. For the LAI calculation a top view image was taken. A standard 100 cm^2^ black square, close to the plant stem, was also present in the image as a reference to calculate the visible leaf area (LA) from the top view. The steps for image processing were the common in this scenario. The RGB images were transformed to binary images by a linear combination of the RGB planes with coefficients: R = −0.884, G = 1.262, B = −0.311, finally followed by the application of Otsu’s method [25] for separating the pixels coming from plants from the rest of pixels of the background. The coefficients were obtained by a genetic algorithm optimization process [26], a method that has proved to perform a better segmentation of the plants than Excess Green coefficients (E × G = 2G − R − B) [27].

#### 2.2.3. RGB-D Microsoft Kinect® v2-Based System

Microsoft Kinect v2 is a motion-sensing device that operates with the principle of the time of flight method for calculating the camera’s distance to a scanned scene. Microsoft Kinect v2 can provide quite detailed information about its surroundings, delivering approximately 2 gigabits of data per second from the surroundings it digitalises. The device is composed of an RGB camera, a depth camera, an infrared (IR) camera and an array of microphones. The RGB camera captures 1080 p video that can be displayed in the same resolution as the viewing screen and can automatically adjust the exposure time in order to create brighter images. Therefore, even though it can capture up to 30 fps (frames per second), the maximum number of frames is limited to 15 fps. The camera creates raw colour images with a 1920 × 1080 resolution. The infrared sensor can track objects without visible light at a resolution of 512 × 424 pixels, allowing the tracking of IR reflective objects while filtering out the IR light. The field of view is wider in the IR camera than the RGB camera 70 degrees horizontally and 60 degrees vertically. The device estimates distances by phase detection to measure the time taken for light to travel from the light source to the object and back to the sensor. Since the speed of light is known, the distance is calculated by the system estimating the received light phase at each pixel with knowledge of the modulation frequency; the operational range is between 0.5 m to 4.5 m. 

The acquisition process was executed using an Intel CORE i7-4710HQ laptop with Windows 7, 16 GB of RAM memory and a 2 GB Nvidia GeForce GTX graphic card with a graphics processing unit (GPU). The raw data were recorded using Kinect Studio in video mode. The plant was scanned according to a concentric track, similarly to the image acquisition in the previous system. The reconstruction software fuses the different consecutive and overlapped depth images. A fully automated process was developed and used in this procedure by overlapping depth images using the algorithm described in [28], with a variant of the iterative closest point (ICP) algorithm [29], which improves the accuracy of the point cloud.

Following image acquisition and Kinect v2 scanning, manual measurements were undertaken in every plant wherein the number of leaves and plant height were determined. The maximum height of each plant was calculated using a cylinder extended from the plant base to the end of its main stem which indicated the maximum stem height [30]. For LA calculation, a leaf area metre, model Li-Cor 3100 (Li-Cor, Lincoln, NE, USA), was used. Thereafter, plants were stored in paper bags, and the samples were dried at 78 °C for 48 h to determine the oven-dry weight of the plant biomass.

#### 2.2.4. Point Cloud Processing

Point clouds are 3D-polygon meshes of the plant geometry; they were created by both systems and methods and processed on the aforementioned laptop. This solid 3D model that was recently processed off line in the open source software Meshlab® (Visual Computing Lab ISTI-CNR, Pisa, Italy), which can manage and plot the 3D outputs previously generated in Agisoft PhotoScan by the self-developed software for Kinect v2 data processing. Meshlab® was used in the processing of the unstructured 3D models using remeshing tools and filters to clean and to smooth the created 3D models. Three steps were used for processing the point clouds. First, noise and outliers were removed from the point cloud by an internal filter, which removed individual points at 0.5 cm outside the grid. Thereafter, the target plant was isolated after removing every source of interference inside the bounding box, i.e., the ground and the rest of objects present in the scene were deleted using colour filters. Due to the presence of green objects in the bounding box, human supervision was needed for the final isolation of the plant. Finally, the extraction of the main parameters that were compared with ground truth data was implemented. Plant height was measured by a measuring tool incorporated to the software and number of leaves in the processed solid were visually counted calculated to be compared with actual plant height, number of leaves and dry biomass.

#### 2.2.5. Statistical Analysis

The information calculated from the final models was compared with the ground truth values measured in the plants. The maximum plant height, leaf area (LA) and number of leaves were calculated from the digital models and later correlated with the actual parameters of height, LA, number of leaves and dry biomass. Pearson´s correlation coefficients were used as a first validation on simple linear regressions and provided an evaluation of the results’ trend between the actual values and those extracted from the model. Then, the root mean square error (RMSE) and the mean absolute percentage error (MAPE) were calculated for the error estimation in order to provide an overview of how well the data fit the model. 

## 3. Results

The plant reconstruction method created different models (Figure 2), which were similar to reality; three-dimensional models were properly reconstructed lacking only small details. These errors varied slightly between both systems the Kinect-based v2 measurements were more imprecise than the photogrammetry measurements. However, in all of the models, the errors appeared at the end of leaves and branch borders. Although the final end-details were not properly reconstructed, the reconstruction displayed high fidelity, as seen by the statistical analysis. The measured parameters using the modeling systems showed a high correlation with the LA and dry biomass ground truth data. Indeed, strong consistency in linear correlation equations was observed between the estimated parameters by the digital 3D model and their actual measured values. Regarding 2D RGB image processing, the results differed with the three-dimensional systems, as their precision was lower in every case. In addition, the number of leaves was always accurately estimated by the three-dimensional methods, while the 2D method always underestimated the number of leaves due to the overlapping of upper leaves. No missed information was found on 3D reconstruction. Every leaf appeared reconstructed in the digital models with only small differences in LA estimation, as shown in Figure 2. Although the general values were similar between both three-dimensional methods, small differences arose when the sampled crop was analyzed independently. Regarding 2D, dicots showed less underestimation of number of leaves. Small leaves on the bottom were occluded by top ones. Only one to two leaves were occluded on sugar beet and sunflower plants. In maize plants due to the parallelism between leaves, the number of occluded leaves varied from 0 (small plants) to a maximum of four true leaves in the case of those plants at BBCH 19. These slight differences can be due to manual movement of the camera and areas becoming hidden while scanning [31]. Therefore, fine details can be improved by increasing the sampling size or reducing the distance between the camera and the target plant. However, this modification in the acquisition process would increase the time required for data acquisition and the data processing. 

While separate analyses concerning each crop showed differences between them, the sampling methodology (SfM, RGB-D or planar RGB) was the main factor in terms of accuracy of the constructed models (Table 1). The average height of the maize plant was 61.17 cm with an R^2^ = 0.998 between the actual height and the measured height for both Kinect-based calculations and the photogrammetry reconstruction methods at a significance of P < 0.01. Similarly, the results of the RMSE with values of 7.58 and 4.48 and MAPE values of 4.34% and 3.08% for each method, respectively, confirmed the accuracy, as they indicate little deviation of the actual values from those estimated by the model. Varied results were found when considering every crop individually; dicot crops were more similar to the real plants than the monocot crop (maize). In addition, the plant growing stage influenced the results obtained, with smaller plants showing slightly higher differences with the ground truth data. Considering the dicotyledonous crops (sunflower and sugar beet), the results improved the accuracy with respect to the maize models- the leaf shape allowed for better reconstruction leading to better results. The actual height of sunflower and sugar beet averaged 34.67 cm and 21.28 cm, respectively. The photogrammetry and Kinect v2 models in sunflower plants generated underestimates of 1.3 cm and 0.94 cm, respectively, which was similar for sugar beet plants wherein underestimates of 0.91 cm for the maximum height in both crops were calculated for the 3D models. Nevertheless, regression analysis showed a strong correlation (R^2^ = 0.998 and R^2^ = 0.999) with extracted values from the model (Figure 3). Both crops’ RMSE values exhibited similar tendency of lower values than maize plants for both methodologies. The comparison of sugar beet models with the actual values reached a RMSE of 0.81 cm for the photogrammetric method, and a similar value was obtained for sunflower the difference with ground truth showed an RMSE of 0.89 cm and a MAPE error of 2.7%. When considering the use of Kinect v2, the calculated values of the RMSE and MAPE showed a similar tendency; these parameters showed that maize plants models had higher error than dicots crops. However, the Kinect v2 models showed slightly higher errors than those created by photogrammetry (Table 1). The underestimation of maximum height in maize plants can be affected by the difficult identification of the plant at the end of the stem. Although this point is clear for dicots species, it remains unclear for maize plants with regard to their leaves and stem. While the measurements were conducted manually in the models, the use of an automated approach of colour segmentation followed by a definition of the region of interest defined by the stem through a circular Hough transform could improve the results [23]. Although, the use of this automated process would increase the accuracy on height measurements, the current level of detail can be enough for most of the phenotyping purposes; the possibilities of both methods to determine height have been displayed. While the acquisition time and post-processing is higher in the case of photogrammetry, slightly better results were obtained in terms of determining plant height. 

Regarding the comparison with ground truth data, the estimation of LA using three-dimensional models, was in good agreement with the reality. The RGB models, which were 2D-based, reached low values of agreement due to leaf overlapping. The actual averaged values of LA were 989.11 cm^2^ for maize, 657.44 cm^2^ for sugar beet and 495.22 cm^2^ for sunflower plants; planar models underestimated these values, and the correlations were not significant. Thus, the use of 2D images for LA determination cannot be considered an accurate method. When maize plants were modelled, the values extracted from the model averaged an LA of 868.3 cm^2^ and 954.4 cm^2^ for the Kinect-based system and the photogrammetry reconstruction methods, respectively. The correlation between actual and model values was significant at P < 0.01 with R^2^ = 0.997 and R^2^ = 0.995, respectively (Figure 4). The statistic parameters of RMSE and MAPE showed high accuracy in model fitness compared to the actual LA. The RMSE values were of 4393.4 cm^2^ and 22267 cm^2^, and MAPE values revealed a deviation of 3.88% and 11.66% for photogrammetry and Kinect v2, respectively. Similar to the aforementioned height values, the dicot crops (sugar beet and sunflower) were modelled with higher accuracy with respect to LA (Table 2). The MAPE values showed a deviation of approximately 1% when plants were modelled with photogrammetry and even higher for Kinect v2 models. The regression models were always significant at P < 0.01 (Figure 4) with low deviations. Similar values were obtained when correlating LA with the dry biomass (Figure 5).

Although there are differences between the studied methods, both of the methods are consistent and generate significant values. The photogrammetry models attained higher fidelity, as they have a better capacity of reconstructing end-details and thinner stems; this effect was more pronounced in maize plants due to their elongated shape. Similar results were obtained by Andújar, et al. [23] when modeling weed plants. Our study showed that there are still some technical aspects to be improved, such as an accurate reconstruction of the end-details, mainly for monocot plants. Additionally, the effect of the underestimation of height underestimation could be corrected with the automated recognition of the region of interest by image-processing software in combination with the reconstruction algorithm, for instance through a colour identification. The use of SfM can be combined with other algorithms for automatic recognition helping breeders to take fast decisions. However, the method requires of automatic system for image acquisition and processing since the time–cost relationship is not suitable for large scanning areas. Although, the use of the current process is suitable for research purposes it needs of a integration on autonomous systems for reducing the time-cost function, reducing the use of human labor during the image acquisition process. Rose, et al. [30], using a similar approach of SfM- and MVS-based photogrammetric methods, created models at high resolution in lab conditions. These authors reported missing parts and triangulation errors at the leaf and branch borders which are similar to the current study. However, the analysis of different crop species has led to differing results mainly because overestimation is due to leaf borders and stem borders. Thus, thinner plants, such as monocots, would lead to more inaccurate results. This effect was also observed for all methods studied; although, there was a lower level of detail, the methods revealed similar effects, usually showing differences regarding plant shape. The close-up laser scanning method has shown a high accuracy in point cloud reconstruction in combination with the steps needed to create a 3D model; this method is fast and attains a high level of detail. However, investments costs are higher and the use of this method requires calibration and warming-up [32]. The use of RGB images in phenotyping processes considerably reduces the acquisition cost. Imagery systems are the most common system and although the method has been highly expanded, the effect of occlusions is challenging and the usage of 2D phenotyping and its usage could be limited for some specific scenarios [33]. Thus, three-dimensional modeling is rapidly expanding, as the higher accuracy of these models leads to better results in plant breeding programmes [34] or in decision-making processes in agriculture [32]. The computational power and the availability of new low-cost sensors favours this development. The two studied methods for 3D reconstruction are both low-budget techniques, which a reach high level of detail. Although, photogrammetry reconstruction leads to more accurate models than RGB-D, the processing and acquisitions cost requires more time. This procedure has shown similar results in other studies; previous approaches using three-dimensional modeling were based on stereovision to reconstruct maize plants, whereby accurate estimations of leaf position and orientation and of the leaf area distribution were undertaken [35]. Additionally, close-range photogrammetry has been used to calculate similar parameters, such as LAI [36], which allows managing the crop needs of pesticide applications by using differential sprayers [37]. The SfM improved the acquisition time by calculating the position of the camera in relation to stereovision [38]. Similarly to our results, a more expensive high-throughput stereo-imaging system for 3D reconstruction of the canopy structure in oilseed rape seedlings showed the valuable options of the SfM for plant breeding programmes [39]. In addition, the easy mode of operation allows the SfM to be implemented in on-field applications. Andújar, et al. [23] reconstructed weed plants by the SfM and MVS to create 3D models of species with contrasting shapes and plant structures. The models showed similar conclusions with good consistency in the correlation equations, as dicot models were more accurate than monocots. Indeed, using a fixed camera position with a tripod increases accuracy. Santos and Oliveira [40] reconstructed basil plants using the SfM; however, these authors concluded that the method was not suitable for very dense canopies. Another study using MVS-modelled plants provided a good representation of the real scene but did not fully reconstruct branch and stem details [41]. Every studied case concluded that measuring leaves and stem is possible through different photogrammetry processes. Completely reconstructing small leaves and thin stems to reach a high level of accuracy requires high cost-time procedures and errors are mostly focused on small leaves and stem borders. Increasing the number of images, lowering the distance from the camera and filling holes in the mesh by manual or automatic processes would lead to better models; these processes, however, will increase the cost of model creation. 

A compromise in the relation between cost and accuracy must be developed. While the cost of photogrammetry processes is low when the SfM is applied, generating more detailed models would increase time and cost. In this case, RGB-D cameras have shown their lower acquisition time and model processing. However, the terminal parts and small details were not properly reconstructed. Thus, the use of this type of camera can be suitable for on-line methods which requires fast decision. The obtained results showed a slighter underestimation of the calculated values than photogrammetry procedure; however, the acquisition time was much lower. Thus, its use in field applications or situations in which high detail is not demanded, such as agronomical plant management at field level can widely reduce the applied inputs, i.e., the use of this models can be helpful to adapt the use of agrochemicals to the calculated plant volume or LA. The low cost of the procedure and the sensing device could help farmers and machinery developers to develop site-specific applications thought this method. On the other hand, the SfM requires almost 40 images per plant along a concentric track to create a proper model, one second of acquisition time was enough to reconstruct plant shape when Kinect v2 measurements were taken, one second of acquisition was enough to reconstruct plant shape. Thus, the models created with this principle lead to a much faster process of data acquisition and processing. However, SfM is a suitable and budget solution for breeders demanding a low-cost system for decision making processes. The possibilities of the use of depth cameras such as Kinect v2 have been previously probed in other scenarios with similar results. The scanning of lettuce plants showed that Kinect-measured height and projected area have fine linear relationships with actual values [42]. Similarly, Hämmerle and Höfle [43] derived crop heights directly from data of full-grown maize and compared the results with LiDAR measurements. The obtained results showed similar values to our study with a general underestimation of crop height, and the combination of more point clouds of Kinect v2 would increase accuracy. However, this approach will increase the acquisition and processing time, and the combination of point clouds would require online approaches for registration [44]. Thus, an improved solution must be developed to reach the final target. The use of the close photogrammetry method could propose a budget solution in breeding programmes or on-field mapping when decision-making processes need not be quickly undertaken. On the other hand, the current algorithms for RGB-D processing allow Kinect v2 measurements to rapidly create models which can be used in phenotyping scenarios when high resolution is not demanded or for on-field applications. 

## 4. Conclusions

The studied methods have shown that deriving crop height or leaf area is possible though three-dimensional modeling, creating models of high value for plant breeding programmes or precision agriculture. These models tend to underestimate crop height and leaf area with differing results, whereby the RGB-D modeling approach created realistic models; however, the underestimation was higher than those models created by the SfM. In addition, the obtained values were influenced by plant structure; the thinner and elongated shape of monocot plants led to lower accuracy in both methods. Since the errors were mainly produced at leaf and stem borders, the elongated shape of this type of plant induced more errors. Regarding the-two dimensional RGB images, leaf area was highly underestimated. Thus, the use of this method for plant phenotyping was not be considered as an appropriate method. The comparison of the studied methods can help users on decision-making process regarding the proper method. The possibilities of the three methods have been shown; choosing a method must be based on the target. Although close range photogrammetry produces highly detailed models, it results in a higher processing time compared to the other methodologies. Increasing computational power would allow rapid model processing that is able to analyze growth dynamics at higher resolutions in the case of photogrammetry or undertaking online field applications using RGB-D systems.

## Figures and Tables

**Figure 1 sensors-19-02883-f001:**
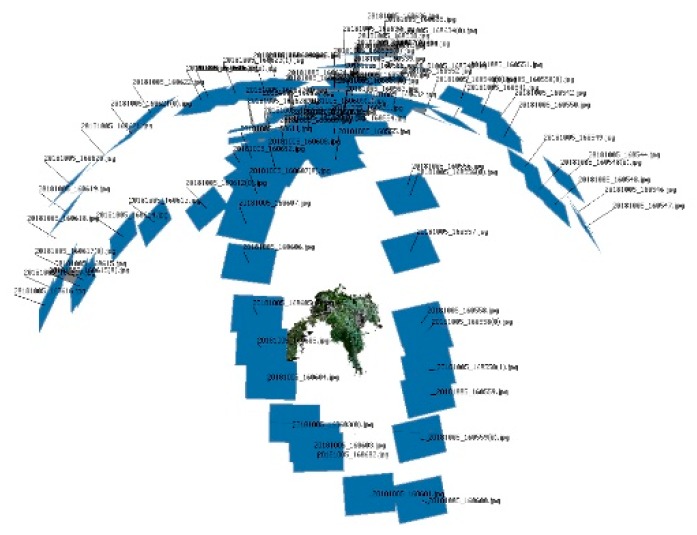
Position of the shoots during image acquisition.

**Figure 2 sensors-19-02883-f002:**
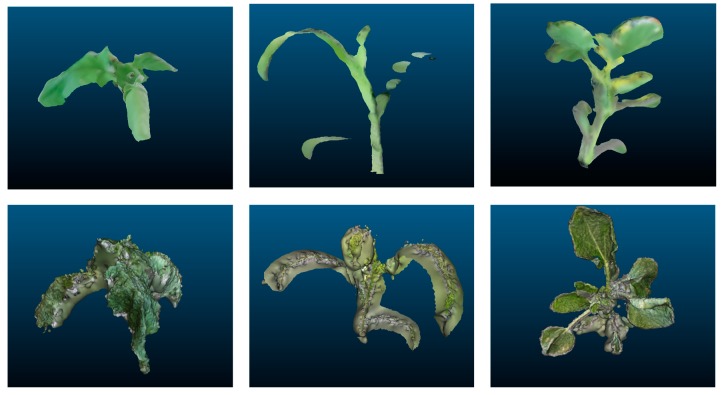
Examples from different perspectives of different crop models reconstructed using Kinect v2 (**top**) and photogrammetry (**bottom**) reconstruction methods.

**Figure 3 sensors-19-02883-f003:**
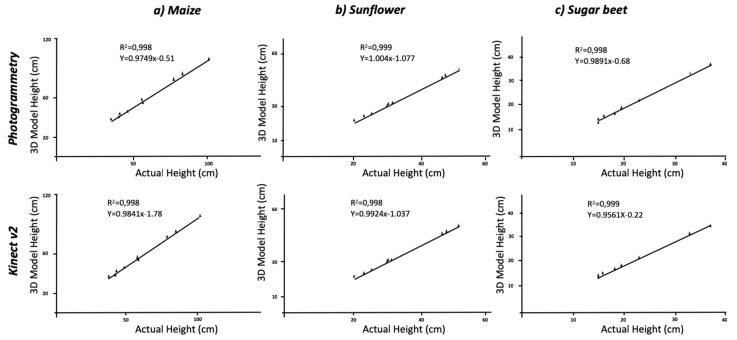
Regression analyses comparing actual plant height versus estimated height using 3D modeling methods.

**Figure 4 sensors-19-02883-f004:**
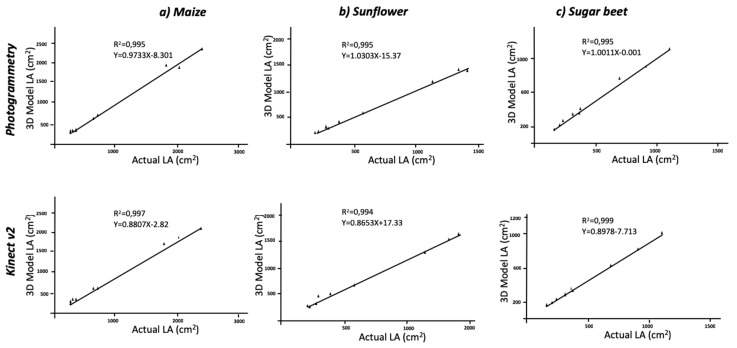
Regression analyses comparing actual leaf area versus estimated leaf area using 3D modeling methods.

**Figure 5 sensors-19-02883-f005:**
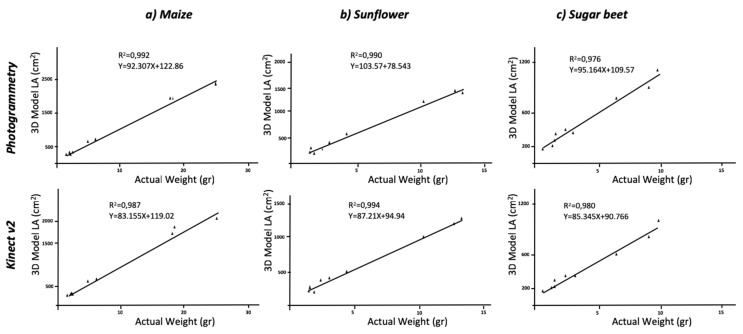
Regression analyses comparing actual leaf area versus measured dry biomass using 3D modeling methods.

**Table 1 sensors-19-02883-t001:** Approximation of time-accuracy for the studied methods. The presented values show an average of the models and the averaged values obtained during the process.

	Distance to plant (cm)	Acquisition time (s)	Processing time (s)	Deviation (mm)
SfM	50	200	1800	2
RGB-D	50	10	30	5
Planar	50	1	1	0

**Table 2 sensors-19-02883-t002:** Root mean square error (RMSE) and mean absolute percentage error (MAPE) calculated values for photogrammetry and Kinect v2 methods in three different crops.

		Photogrammetry			Kinect v2		
		Maize	Sunflower	Sugar Beet	Maize	Sunflower	Sugar Beet
Height (cm)							
RMSE		4.48	0.89	0.81	7.58	1.68	0.91
MAPE		3.08	2.7	4.14	4.34	3.61	3.87
Leaf Area (cm^2^)							
RMSE		4393.4	1175.7	472.7	2,2667.7	9146.7	4033.5
MAPE		3.88	0.8	0.57	11.66	8.31	11.26

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
