# Peer review of "Low-Cost Three-Dimensional Modeling of Crop Plants"

_sensors, 2019, doi:10.3390/s19132883_

Reviewer 1 Report

This paper reported the use of photogrammetry and RGB-D camera to create 3D models for three different crops. Parameters were obtained from the 3D models, and compared to the ground truth. The results seem impressive. However, the methods used by the authors are not new (the 3D modeling pipeline of plants using photogrammetry has been published—reference No.23, as well as the RGB-D camera based pipeline—reference No.19, and many others). More emphasis should be placed on what useful new information this manuscript provides. Other comments below:

1. The manuscript could be improved by adding an analysis of the time-accuracy tradeoff. Does increasing the number of images or decreasing the camera distance improve the accuracy significantly? How long does the acquisition and processing take?

2. No where in the manuscript were the results/data presented for the number of leaves, and 2D RGB method.

3. Closer plants were removed to monitor the target plant. In many field experiments, this is unacceptable. The authors should acknowledge in the manuscript that there are limitations when they made a statement that the methods can be used for field applications.

4. The authors used “modelling” and “modeling” in the abstract/manuscript. Please keep the spelling consistent.

Specific comments:

Line 133: NDVI is not defined. Need to spell out abbreviation on first use.

Line 252-253: How to calculate plant height, number of leaves, and leaf area from  3D models?

Line 285-286: Please elaborate more about the sampling methodology.

Line 424-426: This sentence is unclear to me. RGB-D method created more realistic but less accurate models? What does this mean?

Figure 2: For comparison of the two methods, it would be better to display models reconstructed for the same plant from the same viewing location.

Figure 3-5: need to add units to the axis legends. Additionally, regression equations should be added.

Table 1 should include units.

Author Response

We sincerely thank the reviewers for constructive criticisms and valuable comments, which were of great help in revising the manuscript. Accordingly, the revised manuscript has been systematically improved with new information and additional interpretations. The attached document contains the changes. 

Reviewer 2 Report

First, I am not sure if I understand the contribution of this manuscript. That is, besides presenting a common application of 3D sensing to Ag, the goal of paper is not clear to me. For one, according to the authors, the goal of the paper is "to assess the combination of some of the most novel methods and sensors for plant reconstruction and to compare them to the planar RGB method” (i.e. SfM).  However, some of the so-called "most novel methods" are not that novel anymore.  Also, there are WAY too many details in how those methods were employed -- eg. choice of camera resolution, type and model for the time-of-flight device (in this case a Kinect), distance to objects, algorithms for bundle adjustment, stitching, texturing, etc... -- that were either omitted or not clearly discussed in their choices which could affect the results immensely. In that sense, it is hard to assume that this paper presents a fair or useful comparison between a time-of-flight sensing approach versus a multi-stereopsis-based approach without further discussing these same choices.

Besides, if that was the case of a comparison between two sensing approaches, there are many other papers in the literature that did compare those two methods already.  Now, if this is a comparison of the sensors *under* the target application in Precision Ag, then again, the authors should spend more time justifying the choices and why they represent a fair choice in such context.

Btw, photogrammetry is a horrible term to refer to what is actually dense reconstruction using multi-stereopsis (i.e. many of the algorithms for SfM or SLAM, depending on the context in which they are mentioned -- computer vision or robotics).

Other minor issues include:

The abstract is way too long.

The paper is written reasonably well, but there are some major problems with the writing that makes it very confusing. For ex, the paper presents 2 methods for 3D modeling and 1 for 2D. However, it some points in the text,

they refer to "the 2D methods".  Also, some times the use of "3D method" (i.e. singular) is used without any qualification of which one of them. (eg. line 272).  Finally, the use of the term "3D models" is sometimes used w/o

clear information on 'from which method'. (e.g. line 300)

The authors mention the error at the end of the leaves.  That is a WELL KNOWN problem with TOF sensors, but yet, they seem to have found it in both 3D cases.  Why?!!  Any comments on that?

I don't understand why self occlusion was a problem pointed out or encountered only in the 2D case (line 274-275)

Figure 2 shows VERY different qualitative results for the two 3D methods (e.g. much thicker than normal leaves in the third image in the first row and the level of detail in texture, leaf venation, broken leaves, etc...) but yet, the

authors seem to conclude that the two methods provided similar results (sure! in measured heights, LAI, and such, but still, if this paper is to compare the two methods from a sensors point of view -- as opposed to for the target

application...).

Author Response

We sincerely thank the reviewers for constructive criticisms and valuable comments, which were of great help in revising the manuscript. Accordingly, the revised manuscript has been systematically improved with new information and additional interpretations. The attached document contains the changes. 

Round  2

Reviewer 1 Report

@page { margin: 0.79in } p { margin-bottom: 0.1in; line-height: 120% } a:link { so-language: zxx }

Overall, the revised manuscript seems improved, but the work is still unfocused.  The new paragraph added in the introduction section said “Nevertheless, there are still issues that should be improved and considered during the reconstruction processes. Boundaries and limitations of methods must be clearly stated for its usage. Some aspects that need to be improved, such as the accurate reconstruction of end-details, should be well know before of applying reconstruction techniques for more detailed applications.” It is good that it explains clearly why the authors wants to compare different methods. However, boundaries and limitations of the methods employed here were not shown. And no such discussion about improvement was shown. More explicit information is still needed about the contribution of this manuscript. The authors have to show useful information for others who need to apply reconstruction techniques for their applications.

Author response (1): The minimum number of images was set up by a previous study to determine the lowest number of images for a proper reconstruction. These number of images guarantees a proper reconstruction. An increasement in the images did not provide any improvement in the accuracy and the extracted values did not showed any statistical significance. The method and distance to the target objects was set by [23].

In Lines 449-450, the authors made a statement that “Increasing the number of images or decreasing the camera distance could reduce that error; however, the acquisition and processing time would be increased.” I am confused. Will increasing the number of images help or not?

Author response (2): since the value resulted not significant, some results are avoided in the manuscript and tables

Since the authors discussed the results of the number of leaves, I think the data should be included in the manuscript.

Author Response

We gratefully acknowledge your valuable comment and the time spent on the revision of the manuscript. They have helped to improve the quality of the paper. If any additional review is needed we will keep doing our best to improve the manuscript according to your expertise. Thanks for your hard work in revising our paper. Please check the attached document with the review
